# Sound Randomized Smoothing in Floating-Point Arithmetic

**Václav Voráček, Matthias Hein**
Tübingen AI Center, University of Tübingen

## Abstract

Randomized smoothing is sound when using infinite precision. However, we show that randomized smoothing is no longer sound for limited floating-point precision. We present a simple example where randomized smoothing certifies a radius of $1.26$ around a point, even though there is an adversarial example in the distance $0.8$ and show how this can be abused to give false certificates for CIFAR10. We discuss the implicit assumptions of randomized smoothing and show that they do not apply to generic image classification models whose smoothed versions are commonly certified. In order to overcome this problem, we propose a sound approach to randomized smoothing when using floating-point precision with essentially equal speed for quantized input. It yields sound certificates for image classifiers which for the ones tested so far are very similar to the unsound practice of randomized smoothing. Our only assumption is that we have access to a fair coin.

## 1 Introduction

Shortly after the advent of deep learning, it was observed in Szegedy et al. (2014) that there exist adversarial examples, i.e., small imperceptible modifications of the input which change the decision of the classifier. This property is of major concern in application areas where safety and security are critical such as medical diagnosis or in autonomous driving. To overcome this issue, a lot of different defenses have appeared over the years, but new attacks were proposed and could break these defenses, see, e.g., (Athalye et al., 2018; Croce and Hein, 2020; Tramer et al., 2020; Carlini et al., 2019). The only empirical (i.e., without guarantees) method which seems to work is adversarial training (Goodfellow et al., 2015; Madry et al., 2018) but also there, a lot of defenses turned out to be substantially weaker than originally thought (Croce and Hein, 2020).

Hence, there has been a focus on certified robustness. Here, the aim is to produce certificates assuring no adversarial example exists in a small neighborhood of the original image. For the neighborhood, typically called threat model, one often uses $\ell_p$- balls centered at the original image. However, there also exist other choices, such as Wasserstein balls (Wong et al., 2019; Levine and Feizi, 2020) or balls induced by perceptual metrics (Laidlaw et al., 2021; Voráček and Hein, 2022). The common certification techniques include (1) Bounding the Lipschitz constant of the network, see Hein and Andriushchenko (2017); Li et al. (2019); Trockman and Kolter (2021); Leino et al. (2021); Singla et al. (2022) for the $\ell_2$ threat model and Zhang et al. (2022) for $\ell_\infty$. (2) Overapproximating the threat model by its convex relaxation (admittedly, bounding Lipschitz constant can also be interpreted this way), possibly combined with mixed-integer linear programs or SMT; see, e.g., Katz et al. (2017); Gowal et al. (2018); Wong et al. (2018); Balunovic and Vechev (2020). (3) Randomized smoothing (Lecuyer et al., 2019; Cohen et al., 2019; Salman et al., 2019), which is hitherto the only method scaling to ImageNet. Note that the concept of randomized smoothing may also be interpreted as a special case of (1), see Salman et al. (2019).

All of these certificates expect that calculations can be done with unlimited precision and do not take into account how finite precision arithmetic affects the certificates. For Lipschitz networks (1), the round-off error is of the order of the lowest significant bits of mantissa, which we can estimate to be in the orders of $\sim 10^{-8}$ for single-precision floating-point numbers. Thus, we should assume that the adversary can also inject $\ell_\infty$-perturbation bounded by $\sim 10^{-8}$ in every layer. However, since the networks have small Lipschitz constants by construction, those errors will not be significantly magnified. Although we cannot universally quantify the numerical errors of Lipschitz networks,

they will likely be very small and in particular, can be efficiently traced during the forward pass so that the certificates can be made sound. For the verification methods from category (2), previous works have shown that numerical errors may lead to false certificates for methods based on SMT or mixed-integer linear programming (Jia and Rinard, 2021; Zombori et al., 2021). However, it is possible (and often done in practice) to adapt the verification procedure to be sound w.r.t. floating-point inaccuracies (Singh et al., 2019); thus, the problem is not fundamental, and these verification techniques can be made sound. For randomized smoothing certificates (3), Jin et al. (2022) perform floating-point attacks on certifiably robust networks and indicate the existence of false certificates; see Appendix F for a discussion. The recent work of Lin et al. (2021) focuses on randomized smoothing when using only integer arithmetic in neural networks for embedded devices, so they will, by definition, not have problems with floating-point errors. On the other hand, it does not cover some modern architectures, such as transformers. Furthermore, the way the certificates are computed is derived from the continuous normal distribution; thus, the certificates are approximate, see Appendix G. Another direction is so-called derandomized smoothing - methods that remind randomized smoothing but are deterministic. See, e.g., Levine and Feizi (2021).

In this paper, we make the following contributions[1]:

1. We perform a novel analysis of numerical errors in randomized smoothing approaches when using floating-point arithmetic and identify qualitatively new problems.

2. Building on the observations, we present a simple approach for developing classifiers whose smoothed version will provide fundamentally wrong certificates for chosen points and discuss how this could be exploited in practice.

3. We propose a sound randomized smoothing procedure for floating-point arithmetic with negligible computational overhead for image classification compared to the unsound practice.

While we could not find substantial differences of our sound certificates compared to the unsound practice for our tested classifiers, a lack of a counterexample is not a proof of the correctness. The past has shown that such gaps will be exploited by malicious actors in the future. It is to be expected that certificates of adversarial robustness are required for classifiers used in safety-critical systems (see European AI act European Commission (2021)) and thus will be controlled by regulatory bodies. A malicious company could use then the problems of randomized smoothing in floating-point arithmetic to provide fake certificates on a known/leaked test set. Since our sound randomized smoothing procedure for floating-point arithmetic comes at essentially no additional cost for quantized input e.g., images, we believe that using our sound procedure should always be used for such domains.

**Manuscript organization:** We start with the definition of randomized smoothing in Section 2, then we continue with the introduction of floating-point arithmetic following the IEEE standard 754 (iee, 2008) in Section 3. In Section 4, we exploit the properties of floating-point arithmetic and present a simple classifier producing wrong certificates, and we follow with the identification of the implicit assumptions of randomized smoothing. In Section 5 we conclude the main result by proposing a method of sound randomized smoothing in floating-point arithmetic and provide an experimental comparison of the old unsound and the new sound certificates.

## 2 RANDOMIZED SMOOTHING

Throughout the paper, we consider for clarity the problem of binary classification, but every phenomenon we discuss can be easily transferred to the multiclass setting. We note that the proposed algorithmic fix, see Appendix K, as well as the experiments in A, are done for the multiclass setting.

We first introduce randomized smoothing and define certificates with respect to a norm ball.

**Definition 2.1.** A classifier $F : \mathbb{R}^d \rightarrow \{0, 1\}$ is said to be certifiably robust at point $x \in \mathbb{R}^d$ with radius $r$, w.r.t. norm $\|\cdot\|$ if the correct label at $x$ is $y \in \{0, 1\}$, and $\|x - x'\| \leq r \implies F(x') = y$.

One way to get such a certificate is randomized smoothing (Lecuyer et al., 2019; Cohen et al., 2019; Salman et al., 2019) which we introduce following. We are given a *base classifier* $F : \mathbb{R}^d \rightarrow \{0, 1\}$.

---

[1]Code is available at `https://github.com/vvoracek/Sound-Randomized-Smoothing`

Its smoothed version is $\hat{f}(x) = \mathbb{E}_{\varepsilon \sim \mathcal{N}(0,\sigma^2 I_d)} F(x + \varepsilon)$, and the resulting hard classifier is $\hat{F}(x) = [\![\hat{f}(x) > 0.5]\!]$, where the Iverson bracket $[\![\text{statement}]\!]$ evaluates to 1 if and only if the statement inside holds true. Using the Neyman-Pearson lemma the following result has been shown:

**Theorem 2.2** ((Cohen et al., 2019)). *Let $F$ be a deterministic or random classifier and let $\Phi^{-1}$ be the inverse Gaussian CDF. If*

$$\mathbb{P}_{\varepsilon \sim \mathcal{N}(0,\sigma^2 I)}\big(F(x + \varepsilon) = c_A\big) \geq p_A.$$

*for some $p_A \in (\frac{1}{2}, 1]$, then $\hat{F}(x + \delta) = c_A$ for all $\delta \in \mathbb{R}^d$ with $\|\delta\|_2 < \sigma\,\Phi^{-1}(p_A)$.*

We call in the following $r(x) = \sigma\,\Phi^{-1}(\hat{f}(x))$ the certified $\ell_2$-radius of $\hat{F}$ at $x$.

We note we require that the output of the base classifier $F$ to be independent of previous inputs and outputs. It is easy to construct an $F$ violating this assumption and producing false certificates, e.g., take $F$ that returns 0 in the first $10^6$ calls and 1 afterwards. For the majority of classifiers, it is intractable to evaluate $\hat{f}(x)$ exactly; therefore, random sampling is used to estimate it and thus only a probabilistic certificate is possible where the probability that the certificate holds can be made arbitrarily close to one if one uses more samples or weakens the certificate. Following the literature, we use 100 000 samples to estimate $\hat{f}(x)$ and then lower bound this by $p$ for certifying class 1 (resp. upper bound it for class 0) so that the failure probability, that is when $p > \hat{f}(x)$ (resp. $p < \hat{f}(x)$), is at most 0.001. The value of $p$ can be computed using tail bounds or classical Clopper-Pearson confidence intervals for the binomial distribution. The actual certification procedure is described in Algorithm 1. However, to keep the example below in Listing 1 as simple as possible, we computed $p$ using a simple Hoeffding bound which we derive in Appendix I. Although it produces a weaker certificate, it is still sufficient for the demonstration.

## 3 COMPUTER REPRESENTATION OF FLOATING-POINT NUMBERS

In this section we briefly introduce the floating-point representation and arithmetic according to standard IEEE-754 (iee, 2008). A detailed version with examples and treatment of other precisions can be found in Appendix C where we present examples in a toy, 8−bit, arithmetic. Here, we introduce only single-precision floating-point numbers.

Single-precision floating-point numbers are represented in memory as sequences of bits $x_1 x_2 \ldots x_{32}$. The first bit is a sign bit, the next 8 bits determine the exponent, and the last 23 numbers determine the mansissa. The conversion in normalized form is as follows:

$$(-1)^{x_1} \cdot 2^{\left(\sum_{i=2}^9 x_i \cdot 2^{9-i}\right) - 127} \cdot \left(1 + \sum_{i=10}^{32} x_i \cdot 2^{9-i}\right).$$

We will write the floating-point operations in circles; e.g., $\oplus, \ominus$ instead of $+, -$ to distinguish them from the mathematical ones which do not suffer from rounding errors.

The addition (or analogically subtraction) of two floating-point numbers is performed in three steps. First, the number with the lower exponent is transformed to the higher exponent; then the addition is performed (we assume with infinite precision), and then the result is rounded to fit into the floating-point representation. An example is provided in C.2 in Appendix C.

Thus, it happens that $x \oplus y = x \oplus z$ for any $x$ and some $y \neq z$. Consequently, there will exist some $w$ such that there is no $v$ for which $x \oplus v = w$. This is the main observation that we will built on and is treated in detail in Appendix C in Example C.3.

### 3.1 CONNECTION TO RANDOMIZED SMOOTHING

We have identified some unpleasant properties of floating-point arithmetic that we will exploit in sequel to provide false certificates. In particular, We will try to determine if a given number could be a smoothed version of a specific number or not. The following observations will help us.

In Reiser and Knuth (1975), it is shown that the identity $((x \oplus y) \ominus y) \oplus y = x \oplus y$ holds apart from a single $y$ for any $x$. It is further shown that the identity $(((x \oplus y) \ominus y) \oplus y) \ominus y = (x \oplus y) \ominus y$

holds always true. On the other hand, there is no evidence that the equality $(x \oplus y) \ominus y = x$ should hold. Indeed, consider $x$ to have a lower exponent than $y$. Then during the addition, $x \oplus y$, the low bits of the mantissa of $x$ are lost. Similarly, if $x \oplus y$ has a different exponent than $y$, then a loss of significance may occur during the second rounding. Finally, consider the case where $x = a \ominus y$, then the identity $(x \oplus y) \ominus y = x$ holds.

Our idea is to make the classifier determine if the observed value $x$ could be a smoothed version of $a$. This can be done precisely, but we only approximate this using the previous observation. The reason is that it is sufficient for the demonstration, and the resulting function (introduced in Equation (1) in the next section) will be simple, suggesting that the phenomenon may occur in standard networks.

## 3.2 FLOATING-POINT ISSUES IN THE CONTEXT OF DIFFERENTIAL PRIVACY

Randomized smoothing has been motivated by differential privacy (Lecuyer et al., 2019). In differential privacy it has been shown in the seminal work of Mironov (2012) that the lowest bits of mantissa can serve as a side channel which yields a substantial discrepancy between the theoretical properties of algorithms of differential privacy, and the properties of their naive implementations, see Mironov (2012); Jin et al. (2021); Bichsel et al. (2021). Consequently, revisions of the standard differential privacy mechanisms accounting for the floating-point errors have appeared, see, e.g., Casacuberta et al. (2022); Canonne et al. (2020), and included in the framework OpenDP (Gaboardi et al., 2020).

In our construction, we utilize of the rounding errors of floating-point addition. On a high level, this is similar to what Mironov (2012) exploit. However, their procedure considers Laplacian noise and the example also exploits the Laplace distribution samplers' properties.

## 4 CONSTRUCTION OF CLASSIFIERS WITH FALSE CERTIFICATES

We present an example of a function $F : \mathbb{R} \to \{0, 1\}$ which is prone to giving incorrect certificates via randomized smoothing; the whole "experimental setup" is captured in Listing 1. The example is based on the observation that we are able to determine if a floating-point number $x$ could be a result of floating-point addition $a \oplus n$ where $a$ is known and $n$ is arbitrary. We construct a function $F_a$ whose behavior we analyzed in Subsection 3.1.

$$F_a(x) = [\![(x \ominus a) \oplus a = x]\!]. \tag{1}$$

We take $F_a$ as the base classifier and consider the smoothed classifier $\hat{f}_a$ it induces with $\sigma = 0.5$. It holds that $\hat{f}_a(a) \approx 1$, therefore if we have enough samples, we may obtain a very large certified radius. Specially, in the example considered in Listing 1 with $100\,000$ samples, we can certify a $\ell_2$-radius of 1.26 around point $a = 210/255$, however $0 = \hat{F}_a(0) \neq \hat{F}_a(a) = 1$, and the point 0 is nowhere near the boundary of the certified ball. In the example in 1, we use a simple Hoeffding bound I instead of the standard bounds of Clopper-Pearson. The Clopper-Pearson bounds certify robust radius 1.9.

```python
import numpy as np
from scipy.stats import norm

sigma = 0.5; num_samples = 100000; alpha = 0.001
f = lambda x: (x - 210/255) + 210/255 == x
noise = np.random.randn(num_samples)*sigma

p1 = f(0+noise).sum()/num_samples                    # 0.46
p2 = f(210/255+noise).sum()/num_samples              # 1.0
p = p2-(-np.log(alpha)/num_samples/2)**0.5
r = sigma * norm.ppf(p)                              # 1.26
```

Listing 1: example of an incorrect randomized smoothing certificate

This construction does not rely on the fact that $F_a(a + \varepsilon) = 1$ for $\varepsilon \sim \mathcal{N}(0, \sigma^2)$ with very high probability, it only serves as a striking example. Similarly, we get $0 = \hat{F}_a(0) \neq \hat{F}_a(200/255) = 1$, despite every point $0, 1/255, \ldots, 255/255$ would be class 1 according to the certificate.

### 4.1 Consequences for image classifiers

We stress that the simple construction generalizes to images. For the remainder of the section, we consider $x \in \{0, 1/255, \ldots 255/255\}^d$ to be a vectorized image with e.g., $d = 3 \cdot 32^2 = 3072$ for CIFAR dataset. Indeed, we could employ a function

$$F_{a,i}(x) = [\![(x_i \ominus a) \oplus a = x_i]\!], \tag{2}$$

which takes a vectorized version of an image as an input. Using such function in Listing 1 would certify that any image with intensity $210/255$ at position $i$ is class 1 with robust radius 1.26, while any image with intensity 0 at position $i$ would be classified as 0; a clear contradiction. We take one step further. Consider a function with a parameter $a \in \mathbb{R}^d$:

$$G_a(x) = \min_{i=1}^d [\![(x_i \ominus a_i) \oplus a_i = x_i]\!]. \tag{3}$$

It holds that $\mathbb{E}_{\varepsilon \sim \mathcal{N}(0,1)} G_a(a + \varepsilon) \approx 1$; thus, certifying "arbitrarily" high radius (to be specific, with 100 000 samples it is 3.8115 in $\ell_2$ norm), and $\mathbb{E}_{\varepsilon \sim \mathcal{N}(0,1)} G_a(a' + \varepsilon) < 0.5$ for the vast majority of inputs $a \neq a'$. We tried the following experiment; For every image $a$ in the CIFAR10 test set, we created an image $a'$ by increasing the image intensity of $a$ by $1/255$ at 512 random positions. Then it holds that $\mathbb{E}_{\varepsilon \sim \mathcal{N}(0,1)} G_a(a' + \varepsilon) \leq 0.2$ for every CIFAR image $a$ with high probability, even though $\|a - a'\|_2 < 0.09$.

Following this line of examples, let us introduce the base classifier:

$$H_A(x) = \max_{a \in A} G_a(x) = \max_{a \in A} \min_{i=1}^d [\![(x_i \ominus a_i) \oplus a_i = x_i]\!], \tag{4}$$

where $A$ is a set of images. Therefore, when $A$ is the set of CIFAR10 test set images, then we can certify the robustness of the smoothed version of $H_A$ at every point of the CIFAR10 test set for large radii, even though it is vulnerable even to small random perturbations. We remark that $H_A$ can be implemented with a standard network architecture using only linear layers and ReLU non-linearities. To conclude the examples, we state the findings in the upcoming proposition. Since we introduced the machinery only for binary classification, we treat CIFAR10 as a binary classification dataset. For time reasons, we (as it is common in the context of randomized smoothing) only consider 1000 test images for the upcoming proposition; the first 500 images from the test set of both classes.

**Proposition 4.1.** *There is a classifier with certified robust accuracy* $100\%$ *on the first* $1000$ *CIFAR10 test set images* $X \subset [0, \frac{1}{255}, \ldots, 1]^{3072}$ *(where we define class* 0 *to include classes* $0, 1, 2, 3, 4$ *of CIFAR10 and class* 1 *contains the other classes) with* $\ell_2$-*robust radius of* 3 *and failure probability* $0.001$ *using randomized smoothing certificates, while for every point* $x \in X$ *there is an adversarial example* $x'$ *with* $\|x - x'\|_2 \leq 1$.

The proof can be found in Appendix D The past has shown that loopholes can and will be exploited in the future by malicious actors trying to trick certification agencies e.g. see the diesel scandal where car manufacturers detected the test in a lab to fake significantly better pollution values. As the European AI act requires a certain level of adversarial robustness in safety-critical applications, certification agency are likely to evaluate certified robustness in the future. In order to illustrate the problem, we just sketch how the fake certificates of Proposition 4.1 could be exploited. In fact let $M$ be the classifier described in the proof of Proposition 4.1 and let $m(x) = \mathbb{E}_{\epsilon \sim \mathcal{N}(0, \sigma^2 I_d)} M(x + \epsilon)$ be the smoothed version of $M$. One can see that roughly $m(x) \approx \frac{1}{2}$ if $x \notin N(X)$, where $N(X)$ denotes a small neighborhood of the test set $X$. Given a neural network for image classification a simple way to trick the certification agency, would be a new classifier where one uses the neural network whenever $\delta \leq m(x) \leq 1 - \delta$, e.g. $\delta = 0.1$, and otherwise the classifier $M$ of Proposition 4.1. This classifier would inherit the strong *fake* robustness guarantees on the test set from $M$ but behave like a normal classifier on any other input. We emphasize that this problem is resolved by our fix to randomized smoothing in floating point representation of Section 5 which has negligible computational overhead for image classification.

### 4.2 Implicit assumptions of randomized smoothing

The obvious questions after this negative result are: i) what is the key underlying problem in floating-point arithmetic? ii) what are the implicit assumptions in randomized smoothing?, and iii) how can we fix the problem?

The first assumption of randomized smoothing is that samples from a normal distribution are indeed i.i.d. samples. This is not true for floating-point precision due to the rounding; Thus, the resulting distribution from which we observe samples is uncontrolled, and for certification, we should not rely on it. However, violation of this assumption is not the cause of the wrong certificate in Listing 1.

The intuition behind randomized smoothing is that the distributions $D_1 = \mathcal{N}(x, \sigma^2 I)$ and $D_2 = \mathcal{N}(x + \varepsilon, \sigma^2 I)$ have significant overlap for small values of $\varepsilon$. As a consequence, the smoothed classifier $\hat{f}(x) = \mathbb{E}_{\varepsilon \sim \mathcal{N}(0, \sigma^2 I_d)} F(x + \varepsilon)$ evaluated at $x$ also carries information about its value at points near $x$. However, the following observation will prove this wrong in floating-point arithmetic.

Roughly speaking, the supports of two high dimensional normal distributions appear to be almost disjoint, although in one dimension the overlap may be substantial. To support this claim, We performed the following experiment; given point $a \in \{0, 1/255, \ldots 255/255\}$ and $\sigma > 0$, find a point $b \in \{0, 1/255, \ldots 255/255\}$ such that $|a - b| \leq 2/255$ which minimizes the probability that for an $\varepsilon_1 \sim \mathcal{N}(0, \sigma^2)$ there exists a number $\varepsilon_2$ such that $a \oplus \varepsilon_1 = b \oplus \varepsilon_2$. For example, if $a \geq 5/255$ and $\sigma = 1$, then the minimized probability is less than 0.99, and for the majority of $a \geq 5/255$ it is even smaller. In order to see that the distributions are almost disjoint, consider an image, say from a CIFAR dataset, $a \in \mathbb{R}^{3072}$ which has at least half of its channels with intensities greater than $4/255$. According to the previous observation, we can find an image $a'$ such that $\|a - a'\|_\infty = 2/255$ and that the probability that smoothed $a$ at any (non black) position could be a smoothed version of $a'$ is at most 0.99 (this can be exploited by function $F_{a,i}$ from Equation (2)). Therefore, the probability that a smoothed version of the first image could also be a smoothed version of the second image is at most $0.99^{3072/2} \approx 2 \times 10^{-7}$ (this can be exploited by function $G_a$ from Equation (3)). Thus, when we follow the standard practise and use $10^5$ samples to estimate $\hat{f}(a)$ from base classifier $F$, the chances that at least one of the samples belongs to the distribution from which we sample to estimate $\hat{f}(a')$ is at most in the orders $10^{-2}$. Consequently, without any assumptions on the base classifier $F$, $\hat{f}(a)$ carries almost no information about $\hat{f}(a')$.

### 4.3 POTENTIAL REVISIONS OF RANDOMIZED SMOOTHING

The described experiment exploits the floating-point rounding. The errors are in the order of the least significant bits, which are in the order of $10^{-8}$ for single-precision and $10^{-4}$ for half-precision. Since these numerical errors are not controlled, we should assume that the model is adversarially attacked during smoothing, where the attacker's budget is the possible rounding error, denoted as $\mathcal{B}$; therefore, the smoothing (for certifying class 1) should be performed as:

$$\hat{f}(x) = \mathbb{E}_{\varepsilon \sim \mathcal{N}(0, \sigma^2 I_d)} \min_{\varepsilon_2 \in \mathcal{B}} F(x + \varepsilon + \varepsilon_2).$$

To mitigate this problem, during estimating $\hat{f}(x)$, we should certify $F(x + \varepsilon)$. Although the attacker's budget $\mathcal{B}$ is very small for single accuracy and possibly noticeable for the half accuracy, it is not clear how it should be certified, since in randomized smoothing, there are no assumptions on $F$.

Consider $F$ to be a thresholded classifier $F(x) = [\![f(x) > 0.5]\!]$, where $f$ is a neural network, then we could certify that $f$ is constant in $\mathcal{B}$-neighbourhood of the smoothed image. For generic models, this can be done by either bounding the Lipschitz constant of $f$ (w.r.t. an $\ell_\infty$-like norm), or by propagating a convex relaxation (e.g., IBP) through the network. For smoothing, there are usually used deep models. E.g., Salman et al. (2019) used ResNet110 and ResNet50 for certifying CIFAR10 and ImageNet respectively. The bound on the global Lipschitz constant of a deep network by bounding the operator norms of each layer is thus very weak ($\approx 10^{30} - 10^{130}$, depending on the model) and cannot certify $F(x + \varepsilon)$ even under such a weak threat model as the rounding errors in $\mathcal{B}$.

A possible defense against this problem would be to round the input on a significantly larger scale than $\mathcal{B}$ before evaluating $F$. Let the rounding be performed by a mapping $g$, then we would in fact smooth a classifier $F \circ g$. If we consider $\mathcal{B}$ to be in the orders of $10^{-8}$ and we would round it to orders $10^{-2}$, then the probability that $x + \varepsilon$ will be close to the boundary of rounding, i.e., $\exists \varepsilon_2 \in \mathcal{B} : g(x + \varepsilon + \varepsilon_2) \neq g(x + \varepsilon)$ would be on the order of $10^{-6}$, which is then the probability that the attack within the threat model $\mathcal{B}$ could indeed change the input of $F$ at a single position. Consequently, the probability that there is no $\varepsilon_2 \in \mathcal{B}$ which would change the result of rounding is very roughly $\approx (1 - 10^6)^{3072} \approx 0.997$ for CIFAR and $\approx (1 - 10^6)^{150528} \approx 0.86$ for ImageNet. This means that for approximately $86\%$ of the smoothed ImageNet images we can guarantee that

$F(x + \varepsilon + \varepsilon_2) = F(x + \varepsilon)$ and for the others, we could e.g., set $\min_{\varepsilon_2 \in \mathcal{B}} F(x + \varepsilon + \varepsilon_2) = 0$. This replacement of $F$ by $F \circ g$ during smoothing seem to solve the problem for CIFAR and partially also for ImageNet for single precision. For half precision, the problem will persist.

However, even if this adjustment solved the problem with numerical errors satisfactorily during the addition of noise to images, the certificate will still not be sound because we are unlikely to control the normal distribution sampler's performance.The normal distribution samplers implementations used in standard software libraries (e.g., Ziggurat algorithm; Box-Muller transform) transform i.i.d. uniform distribution samples to i.i.d. normal distribution samples. While this is true in theory for unlimited precision; here, we perform floating-point operations. Therefore, the sampling is subject to floating-point errors and we don't observe the actual rounded samples from normal distribution.

While we do not present any example exploiting the subtle errors of the normal distribution samplers, relying on them only keeps a possible loophole in the procedure. As our goal is to propose a sound randomized smoothing procedure in floating-point arithmetic, our work would be incomplete if we addressed only some potential causes of floating-point errors and not the others, even though we think they are harder to exploit. In particular, we note that Mironov (2012) exploited the (standard) floating-point implementation of the Laplace distribution sampler in order to attack the guarantees of differential privacy.

## 5    SOUND RANDOMIZED SMOOTHING FOR FLOATING-POINT ARITHMETIC

In this section, we will derive a sound randomized smoothing certification procedure for floating-point arithmetic. Our only assumption is the access to i.i.d. samples of a fair coin toss, which is equivalent to having access to samples from the uniform distribution on integers $0, \ldots, 2^n - 1$ for some $n$. Thus, we assume to have access to uniform samples from numbers representable by `Long` datatype, that is when $n = 64$. We further consider classification tasks where the input is quantized as it is true for images. Throughout the section, we consider the input space to be $\{0, 1, \ldots, 255\}^d$ in order to have clear notation. The generalization to other forms of quantized inputs is generic, but the generalization to real-valued inputs is a bit more involved; we move the discussion to Appendix E. The resulting algorithm is captured in Appendix K in Algorithm 2.

### 5.1    CERTIFICATION OF QUANTIZED INPUT

As discussed in the previous section, it is appealing to quantize the smoothed images before feeding them into the network. Thus, we prepend a mapping $g_k : \mathbb{R}^d \to \{-k, -k+1, \ldots, k+255\}$, for some positive integer $k$ which rounds the input to the nearest integer from its range before the function to be smoothed $F$; therefore, the smoothed classifier (with base classifier $F \circ g_k$) is defined as:

$$\hat{f}(x) = \mathbb{E}_{\varepsilon \sim \mathcal{N}(0, \sigma^2 I_d)} F(g_k(x + \varepsilon)),$$

which we further equivalently rewrite as

$$\hat{f}(x) = \mathbb{E}_{t \sim g_k(x+\varepsilon),\, \varepsilon \sim \mathcal{N}(0, \sigma^2 I_d)} F(t).$$

This treatment is crucial for the method. Instead of adding noise to the input, which is subject to rounding errors, we sample the noised input directly.

### 5.2    DISCRETIZED NORMAL DISTRIBUTION

It remains to show how to obtain i.i.d. samples from the discretized normal distribution

$$\mathcal{N}_D^k(x, \sigma^2) = g_k(x + \varepsilon), \quad \varepsilon \sim \mathcal{N}(0, \sigma^2).$$

We note that the discretized normal distribution is different from the discrete Gaussian distribution used in the context of differential privacy (Canonne et al., 2020).

As discussed in the two final paragraphs of Subsection 4.3, it is not enough to round samples from normal distribution since we cannot guarantee the correctness of the normal sampler. The key observation is that we do not even need to obtain samples from $\mathcal{N}(0, \sigma^2)$ anymore. The resulting distribution from which we want to sample now is discrete. Concretely, we have

Table 1: Time comparison of certification times per image of the standard randomized smoothing and the proposed sound procedure with and without reusing noise. Details in Appendix A.1. For CIFAR10 we used 100 000 random samples and for ImageNet 10 000.

| dataset | standard | proposed w/ reusing | proposed w/o reusing |
|---|---|---|---|
| CIFAR10 (ResNet-110) | 9.82 s | 9.87 s | 31.70 s |
| ImageNet (ResNet-50) | 8.65 s | 8.67 s | 129 s |

$$\mathbb{P}_{t \sim \mathcal{N}_D^k(x,\sigma^2)}[\![t = a]\!] = \begin{cases} \int_{-\infty}^{-k+\frac{1}{2}} \frac{1}{\sqrt{2\pi\sigma^2}} \, e^{-\frac{(x-u)^2}{2\sigma^2}} \, du & \text{if } a = -k, \\ \int_{a-\frac{1}{2}}^{a+\frac{1}{2}} \frac{1}{\sqrt{2\pi\sigma^2}} e^{-\frac{(x-u)^2}{2\sigma^2}} \, du & \text{if } -k < a < k+255, a \in \mathbb{Z} \\ \int_{k+255-\frac{1}{2}}^{\infty} \frac{1}{\sqrt{2\pi\sigma^2}} e^{-\frac{(x-u)^2}{2\sigma^2}} \, du & \text{if } a = k+255. \\ 0 & \text{otherwise} \end{cases}$$

Additionally, the following well-known property of normal distribution holds for the discretized normal distribution as well.

**Proposition 5.1.** *Let $k$ be a positive integer and $x \in \{0, 1, \ldots, 255\}$, then it holds that $\mathcal{N}_D^k(x, \sigma^2) = \max\{-k, \min\{k + 255, t' + x\}\}$, $t' \sim \mathcal{N}_D^{k+255}(0, \sigma^2)$.*

Thus, it is enough to have a sampler from $\mathcal{N}_D^{k+255}(0, \sigma^2)$. The value of $k$ is chosen such that the vast majority of samples from $\mathcal{N}(x, \sigma^2)$ falls into the interval $[-k, k + 255]$. The choice of $k$ does not affect the correctness of the certificates, but may affect the accuracy. In the experiments we chose $k = 6\sigma_{max} = 6$ for inputs from $[0, 1]^d$, so it corresponds to $k = 6 \cdot 255$ in the notation of this section.

### 5.2.1 Discretized normal distribution sampler

Let us denote the quantile function (inverse cdf) of $\mathcal{N}_D^k(0, \sigma^2)$ as $\Phi_{D,k}^{-1}$, then $\Phi_{D,k}^{-1}$ transforms i.i.d. samples from the uniform distribution on the interval $[0, 1) =: \mathcal{U}(0, 1)$ to i.i.d. samples from distribution $\mathcal{N}_D^k(0, \sigma^2)$. To sample from $\mathcal{N}_D^k(0, \sigma^2)$, we first approximate the samples from $\mathcal{U}(0, 1)$ by samples from the discrete uniform distribution on $\{0, \ldots, 2^n - 1\}$ which we will interpret as uniform distribution on $\{0, \frac{1}{2^n}, \ldots, \frac{2^n-1}{2^n}\} =: \mathbf{U}(0, 1)$. Then we only need to compute the $2k + 255$ probabilities with high enough accuracy that we can claim the correctness of $\Phi_{D,k}^{-1}(u)$ for $u \in \{0, \frac{1}{2^n}, \ldots, \frac{2^n-1}{2^n}\}$. This is ensured by using symbolic mathematical libraries allowing computations in arbitrary precision.

Since the distribution $\mathcal{N}_D^k(0, \sigma^2)$ is supported on $2k + 256$ events, there will be $2k + 255$ points $x \in \{0, \frac{1}{2^n}, \ldots, \frac{2^n-1}{2^n}\}$ such that $\Phi_{D,k}^{-1}(x) \neq \Phi_{D,k}^{-1}\left(x + \frac{1}{2^n}\right)$. Now, consider the mapping between samples from $\mathcal{U}(0, 1)$ and $\mathbf{U}(0, 1)$ which rounds down a sample $u$ from the continuous real interval $[0, 1)$ to the closest point $v$ from the set $\{0, \frac{1}{2^n}, \ldots, \frac{2^n-1}{2^n}\}$. Then it holds for the probability $\Phi_{D,k}^{-1}(u) \neq \Phi_{D,k}^{-1}(v) \leq \frac{255+2k}{2^n} \leq 2^{12-n}$ for a choice $k = 7.5 \times 255$. The probability that a produced sample is not the actual i.i.d. sample is thus at most $2^{12-n}$ at one position. Therefore, the probability that all the smoothed images, considering ImageNet sized images with shape $3 \times 224 \times 224$, out of 100 000 smoothed samples are indeed the correct i.i.d. samples from discrete normal distribution is at least $1 - 2^{46-n} > 0.999996$ for $n = 64$.

Therefore, the probability of receiving a sample that might not be the actual i.i.d. sample is negligible. Still, we can check if we receive such a potentially flawed sample $x$ and in that case, we would set $F(x) = 0$ when certifying class 1 (resp. $F(x) = 1$ when certifying 0) for that particular sample.

### 5.2.2 Sampling speed of discretized normal distribution

The sampling is slightly more expensive since we need to threshold the observed uniform samples; however, this is only an implementation issue. On the other hand, it is sufficient to sample i.i.d. noise for just one image 100 000 times and reuse it for all the other images. The certificates will be valid, only the case of failure for different images will not be independent, but it is not required in the

Table 2: Certified radii for a model $F$ smoothed with $\mathcal{N}(0, \sigma^2 I)$ on CIFAR10 test set. Evaluated on $500$ images from the test set highlighting the differences. The model is ResNet-110 taken from (Salman et al., 2019), more details in Appendix A.

Sound smoothing of $F \circ g_k$ via Algorithm 2 for $k = 6$

| certified radius | 0 | 0.1 | 0.25 | 0.5 | 0.75 | 1 | 1.25 | 1.5 | 2.0 |
|---|---|---|---|---|---|---|---|---|---|
| $\sigma = 0.12$ | 0.878 | 0.848 | 0.778 | 0.000 | 0.000 | 0.000 | 0.000 | 0.000 | 0.000 |
| $\sigma = 0.25$ | 0.836 | 0.808 | 0.746 | 0.600 | 0.466 | 0.000 | 0.000 | 0.000 | 0.000 |
| $\sigma = 0.5$ | 0.708 | 0.672 | 0.618 | 0.502 | 0.410 | 0.338 | 0.248 | 0.174 | 0.000 |
| $\sigma = 1$ | 0.512 | 0.492 | 0.448 | 0.380 | 0.316 | 0.278 | 0.230 | 0.182 | 0.112 |

Standard smoothing of $F$ via Algorithm 1

| certified radius | 0 | 0.1 | 0.25 | 0.5 | 0.75 | 1 | 1.25 | 1.5 | 2.0 |
|---|---|---|---|---|---|---|---|---|---|
| $\sigma = 0.12$ | 0.880 | 0.848 | 0.778 | 0.000 | 0.000 | 0.000 | 0.000 | 0.000 | 0.000 |
| $\sigma = 0.25$ | 0.836 | 0.808 | 0.746 | 0.602 | 0.468 | 0.000 | 0.000 | 0.000 | 0.000 |
| $\sigma = 0.5$ | 0.706 | 0.672 | 0.618 | 0.502 | 0.408 | 0.338 | 0.248 | 0.174 | 0.000 |
| $\sigma = 1$ | 0.516 | 0.492 | 0.448 | 0.378 | 0.316 | 0.278 | 0.230 | 0.182 | 0.110 |

literature. As we sample just once for the whole data set, the time spent for sampling is negligible, see Table 1. We discuss the timing in detail in Appendix A.1.

Finally, we wrap up the observations in the following corollary:

**Corollary 5.2.** *Let $F : \mathbb{R}^d \to \{0, 1\}$ be a deterministic or a random function and $g_k : \mathbb{R}^d \to \{\frac{-k}{255}, \frac{-k+1}{255}, \dots \frac{k+255}{255}\}$ maps input to the closest point of its range, breaking ties arbitrarily. Then the following two functions are identical:*

$$\hat{f}_1(x) = \mathbb{E}_{\varepsilon \sim \mathcal{N}(0, \sigma^2 I_d)} F(g_k(x + \varepsilon)), \qquad \hat{f}(x) = \mathbb{E}_{t \sim \mathcal{N}_D^k(x, \sigma^2 I_d)} F(t).$$

*Therefore, to certify $\hat{f}_1$ with base classifier $F \circ g_k$ using randomized smoothing, we can estimate the value of $\hat{f}$ and use it for the certification. Furthermore we can get i.i.d. samples from $t \sim \mathcal{N}_D^k(x, \sigma^2 I_d)$ with arbitrarily high precision using exact arithmetic; thus, the certificate is sound.*

*Remark* 5.3. The empirical performance of the sound and unsound versions of randomized smoothing are essentially equivalent in practice; see Table 2 and also Appendix A for the evidence. However, it no longer incorrectly certifies the example from Listing 1, where it only certifies a radius $0.6$, and the points are distant $0.82$ from each other. Similarly, the smoothed classifier of $M$ from Proposition 4.1 does not contain the universal adversarial perturbations in the certified balls around the points. See Appendix H for more details.

To summarize: we showed how to replace sampling from the normal distribution, where one cannot trace the numerical errors, by sampling from the uniform distribution on integers, where we can bound the failure probability in order to obtain high probability estimates of the output of a smoothed classifier with a prepended rounding mapping. See Algorithms 1, 2 for the comparison of the standard and the proposed certification procedure. We also provide an empirical comparison of the methods in Table 2 and in Appendix A.

# 6 CONCLUSION

In the paper, we described multiple simple ways how to construct models that will be certifiably robust for points of our choice using the standard randomized smoothing certification procedure, although there will be adversarial examples in their close neighborhood.

Most importantly, we provided a sound way to do randomized smoothing in floating point representation which comes at negligible cost in image classification.

## ACKNOWLEDGEMENTS

The authors thank the anonymous reviewers for their comments, which helped improve the quality of the manuscript. The authors acknowledge support from the DFG Cluster of Excellence "Machine Learning – New Perspectives for Science", EXC 2064/1, project number 390727645 and the Carl Zeiss Foundation in the project "Certification and Foundations of Safe Machine Learning Systems in Healthcare". The authors are thankful for the support of Open Philanthropy.

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

# A    EXPERIMENTS

To run the experiments, we used the publicly available codebase of Salman et al. (2019) which is distributed under MIT licence. Our modifications will be publicly available under MIT licence. The experiments were run on a single Tesla V100 GPU. The models we evaluated were chosen arbitrarily from the models Salman et al. (2019) provide in their repository. Their identifications are:

pretrained_models/cifar10/finetune_cifar_from_imagenetPGD2steps/PGD_10steps_30epochs_multinoise/2-multitrain/eps_64/cifar10/resnet110/noise_$\sigma$/checkpoint.pth.tar,

pretrained_models/cifar10/PGD_4steps/eps_255/cifar10/resnet110/noise_$\sigma$/checkpoint.pth.tar

pretrained_models/cifar10/PGD_4steps/eps_512/cifar10/resnet110/noise_$\sigma$/checkpoint.pth.tar

where $\sigma \in \{0.12, 0.25, 0.50, 1.00\}$ for tables 2, 3 and 4 respectively. In Table 2, 100 000 samples are used, whereas for Tables 3, 4 we used only 10 000 samples to evaluate the smoothed classifier.

For Imagenet experiments, we used models:

pretrained_models/imagenet/replication/resnet50/noise_$\sigma$/checkpoint.pth.tar,
pretrained_models/imagenet/DNN_2steps/imagenet/eps_512/resnet50/noise_$\sigma$/checkpoint.pth.tar

where $\sigma \in \{0.25, 0.50, 1.00\}$ for tables 5 and 6 respectively. Again, we used 10 000 samples to evaluate the smoothed classifier.

## A.1    SPEED

The speed is essentially equal for both of the methods, described in Algorithm 1 and 2 respectively because we compute the noise beforehand and then we can use the same set of $n$ noises for every image, where $n$ is the number of samples used to evaluate a smoothed classifier. The time needed to generate the noise is in the order of minutes; thus, negligible compared to the time needed to run the experiments.

To be more precise; we run the experiment on a GPU Tesla V100. For CIFAR10, The (standard) time per image for 100 000 samples used to evaluate the classifier is $9.82 \pm 0.05$s. If we precompute the noise batches (batch size 1000, thus we have 100 batches) and save them to files. Then with every image and every batch we load the corresponding noise, the time is then $10.47 \pm 0.03$s per image. The advantage of this approach is that the change of the codebase is minimal. In our case, we changed two lines of code of Salman et al. (2019) and added one class. The disadvantage is that we do a lot of unnecessary work by loading the same batch of noises multiple times. Finally, if we compute a batch of noises and evaluate the classifier for every test-set point with this noise before sampling a new batch, we get to average time $9.87$s If we compute noises for every image separately, the per image time is $31.70 \pm 0.15$. The $\pm$ denotes standard deviation of time per image. Thus, it is not applicable for the second to last last case. The reported times are from experimental setup of 2 with $\sigma = 0.12$, $k = 1$ (ResNet110). For these experiments, we used (vectorized) procedure analogical to the one in Algorithm 2. It takes 4 minutes to compute the breaking points with SymPy library using exact arithmetic evaluated with sufficient precision (on a single core). Here, we assumed that `torch.randint` produces i.i.d. samples.

For ImageNet, we used the model from Table 5 with $\sigma = 0.5$, $k = 3$ (ResNet50). We used batch size 100, 10 000 smoothed versions to evaluate the per-image times follow:. The time of the standard method $8.65 \pm 0.07$s, the time of the reloading reuse is $12.36 \pm 0.07$s. The time when we sample noise and evaluate every image on that noise is $8.67$s. New noises for every image yields $129 \pm 1.29$s. The time to compute breaking points is about 6 minutes (again, single core).

Table 3: Certified radii for a model $F$ smoothed with $\mathcal{N}(0, \sigma^2 I)$ on CIFAR10 test set. Evaluated on $500$ images from the test set highlighting the differences. The model is ResNet-110 taken from (Salman et al., 2019). See Appendix A for the details.

Sound smoothing of $F \circ g_k$ via Algorithm 2 for $k = 6$

| certified radius | 0 | 0.1 | 0.25 | 0.5 | 0.75 | 1 | 1.25 | 1.5 | 2.0 |
|---|---|---|---|---|---|---|---|---|---|
| $\sigma = 0.12$ | 0.714 | 0.678 | 0.630 | 0.000 | 0.000 | 0.000 | 0.000 | 0.000 | 0.000 |
| $\sigma = 0.25$ | 0.660 | 0.636 | 0.590 | 0.526 | 0.444 | 0.000 | 0.000 | 0.000 | 0.000 |
| $\sigma = 0.50$ | 0.554 | 0.538 | 0.500 | 0.442 | 0.386 | 0.340 | 0.284 | 0.204 | 0.000 |
| $\sigma = 1$ | 0.440 | 0.428 | 0.402 | 0.368 | 0.332 | 0.292 | 0.248 | 0.202 | 0.160 |

Standard smoothing of $F$ via Algorithm 1

| certified radius | 0 | 0.1 | 0.25 | 0.5 | 0.75 | 1 | 1.25 | 1.5 | 2.0 |
|---|---|---|---|---|---|---|---|---|---|
| $\sigma = 0.12$ | 0.712 | 0.678 | 0.630 | 0.000 | 0.000 | 0.000 | 0.000 | 0.000 | 0.000 |
| $\sigma = 0.25$ | 0.664 | 0.638 | 0.592 | 0.522 | 0.444 | 0.000 | 0.000 | 0.000 | 0.000 |
| $\sigma = 0.50$ | 0.556 | 0.540 | 0.500 | 0.440 | 0.386 | 0.338 | 0.284 | 0.194 | 0.000 |
| $\sigma = 1$ | 0.440 | 0.428 | 0.402 | 0.366 | 0.330 | 0.290 | 0.248 | 0.204 | 0.164 |

Table 4: Certified radii for a model $F$ smoothed with $\mathcal{N}(0, \sigma^2 I)$ on CIFAR10 test set. Evaluated on $500$ images from the test set highlighting the differences. The model is ResNet-110 taken from (Salman et al., 2019). See Appendix A for the details.

Sound smoothing of $F \circ g_k$ via Algorithm 2 for $k = 6$

| certified radius | 0 | 0.1 | 0.25 | 0.5 | 0.75 | 1 | 1.25 | 1.5 | 2.0 |
|---|---|---|---|---|---|---|---|---|---|
| $\sigma = 0.12$ | 0.560 | 0.544 | 0.522 | 0.000 | 0.000 | 0.000 | 0.000 | 0.000 | 0.000 |
| $\sigma = 0.25$ | 0.534 | 0.514 | 0.492 | 0.450 | 0.408 | 0.000 | 0.000 | 0.000 | 0.000 |
| $\sigma = 0.50$ | 0.466 | 0.458 | 0.440 | 0.414 | 0.378 | 0.342 | 0.306 | 0.258 | 0.000 |
| $\sigma = 1$ | 0.370 | 0.364 | 0.342 | 0.320 | 0.298 | 0.276 | 0.252 | 0.226 | 0.166 |

Standard smoothing of $F$ via Algorithm 1

| certified radius | 0 | 0.1 | 0.25 | 0.5 | 0.75 | 1 | 1.25 | 1.5 | 2.0 |
|---|---|---|---|---|---|---|---|---|---|
| $\sigma = 0.12$ | 0.558 | 0.544 | 0.522 | 0.000 | 0.000 | 0.000 | 0.000 | 0.000 | 0.000 |
| $\sigma = 0.25$ | 0.534 | 0.514 | 0.492 | 0.450 | 0.410 | 0.000 | 0.000 | 0.000 | 0.000 |
| $\sigma = 0.50$ | 0.464 | 0.458 | 0.440 | 0.414 | 0.380 | 0.340 | 0.308 | 0.264 | 0.000 |
| $\sigma = 1$ | 0.372 | 0.364 | 0.344 | 0.318 | 0.298 | 0.274 | 0.250 | 0.222 | 0.168 |

Table 5: Certified radii for a model $F$ smoothed with $\mathcal{N}(0, \sigma^2 I)$ on Imagenet test set. Evaluated on 1000 images from the test set highlighting the differences. The modelis ResNet-50 taken from (Salman et al., 2019). See Appendix A for the details.

Sound smoothing of $F \circ g_k$ via Algorithm 2 for $k = 12$

| certified radius | 0 | 0.1 | 0.25 | 0.5 | 0.75 | 1 | 1.25 | 1.5 | 2.0 |
|---|---|---|---|---|---|---|---|---|---|
| $\sigma = 0.25$ | 0.661 | 0.636 | 0.614 | 0.559 | 0.498 | 0.000 | 0.000 | 0.000 | 0.000 |
| $\sigma = 0.50$ | 0.597 | 0.586 | 0.549 | 0.509 | 0.460 | 0.428 | 0.383 | 0.330 | 0.000 |
| $\sigma = 1$ | 0.447 | 0.438 | 0.424 | 0.390 | 0.365 | 0.344 | 0.319 | 0.299 | 0.238 |

Standard smoothing of $F$ via Algorithm 1

| certified radius | 0 | 0.1 | 0.25 | 0.5 | 0.75 | 1 | 1.25 | 1.5 | 2.0 |
|---|---|---|---|---|---|---|---|---|---|
| $\sigma = 0.25$ | 0.660 | 0.635 | 0.614 | 0.559 | 0.497 | 0.000 | 0.000 | 0.000 | 0.000 |
| $\sigma = 0.50$ | 0.598 | 0.584 | 0.548 | 0.507 | 0.459 | 0.429 | 0.385 | 0.323 | 0.000 |
| $\sigma = 1$ | 0.447 | 0.439 | 0.424 | 0.390 | 0.365 | 0.344 | 0.320 | 0.297 | 0.240 |

Table 6: Certified radii for a model $F$ smoothed with $\mathcal{N}(0, \sigma^2 I)$ on Imagenet test set. Evaluated on 1000 images from the test set highlighting the differences. The model is ResNet-50 taken from (Salman et al., 2019). See Appendix A for the details.

Sound smoothing of $F \circ g_k$ via Algorithm 2 for $k = 12$

| certified radius | 0 | 0.1 | 0.25 | 0.5 | 0.75 | 1 | 1.25 | 1.5 | 2.0 |
|---|---|---|---|---|---|---|---|---|---|
| $\sigma = 0.25$ | 0.672 | 0.642 | 0.592 | 0.505 | 0.393 | 0.000 | 0.000 | 0.000 | 0.000 |
| $\sigma = 0.50$ | 0.580 | 0.566 | 0.534 | 0.484 | 0.425 | 0.378 | 0.331 | 0.268 | 0.000 |
| $\sigma = 1$ | 0.448 | 0.439 | 0.416 | 0.379 | 0.348 | 0.327 | 0.299 | 0.266 | 0.210 |

Standard smoothing of $F$ via Algorithm 1

| certified radius | 0 | 0.1 | 0.25 | 0.5 | 0.75 | 1 | 1.25 | 1.5 | 2.0 |
|---|---|---|---|---|---|---|---|---|---|
| $\sigma = 0.25$ | 0.672 | 0.641 | 0.593 | 0.503 | 0.389 | 0.000 | 0.000 | 0.000 | 0.000 |
| $\sigma = 0.50$ | 0.581 | 0.564 | 0.534 | 0.486 | 0.423 | 0.377 | 0.337 | 0.270 | 0.000 |
| $\sigma = 1$ | 0.449 | 0.440 | 0.418 | 0.380 | 0.349 | 0.324 | 0.299 | 0.268 | 0.211 |

## B  PROOF OF PROPOSITION 5.1

Let us inspect the probability of observing some $a \in \{-k+1, \ldots, k+254\}$. In that case $\mathbb{P}_{t \sim \mathcal{N}_D^k(x, \sigma^2)}[\![t = a]\!] = \int_{a-0.5}^{a+0.5} \frac{1}{\sqrt{2\pi\sigma^2}} e^{-\frac{(x-u)^2}{2\sigma^2}} du$. For the other distribution it holds that $t' = a - x$ and $\mathbb{P}_{t \sim \mathcal{N}_D^k(0, \sigma^2)}[\![t = a - x]\!] = \int_{a-x-0.5}^{a-x+0.5} \frac{1}{\sqrt{2\pi\sigma^2}} e^{-\frac{u^2}{2\sigma^2}} du$ and the change of the variable $u \to v - x$ concludes the proof of this case. The other two cases are analogical. $\qquad\square$

## C  FLOATING-POINT NUMBERS

In this appendix, we introduce the floating-point representations and arithmetic according to standard IEEE-754 (iee, 2008). For the sake of clarity, in this section, we use 8-bit floating-point number representation instead of the usual $16, 32, 64$ bits, respectively for half, single, and double precision. This appendix is self-contained and repeats the contains also (in a more detailed way) the content presented in the main paper.

Floating-point numbers are represented in memory using three different sequences of bits. The split is $1/3/4$ for the 8-bit example, $1/5/10$ for half-precision, $1/8/23$ for standard single precision, and $1/11/52$ for double precision. The modern GPUs use most often single-precision floating point numbers. We will represent the binary numbers as binary strings of $8$ numbers and start with an example translating binary floating-point representation to the standard decimal one.

**Example C.1.** Consider the binary number 1110 1010. The *first bit is the sign bit*. The number is negative iff the bit is set to 1. In our example, the bit is 1; thus, the number is negative. The *next 3 bits (110) determine the exponent*. It is the integer value of this encoding minus 3, thus, in our example, the exponent is $6 - 3 = 3$. The subtraction of 3 enables that one can represent exponents $-3, -2, \ldots, 4$. The *last sequence is called mantissa* and encodes the number after the decimal point. There is also an implicit (not written) 1 before it. This is a so-called normalized form. Thus, the encoded value of the mantissa is $1.1010$ in binary representation, which is $1 + 1 \cdot 0.5 + 0 \cdot 0.25 + 1 \cdot 0.125 + 0 \cdot 0.0625 = 1.625$ in base 10. The represented number is thus $-1.625 \cdot 2^3 = -13$ in base 10.

### C.1  SUBNORMAL NUMBERS, NANS AND INFS

We note that the introduced floating-point representation is not able to represent $0$ and the smallest representable positive number is 0000 0000 which is $0.125$ in base 10. To represent even smaller numbers, there are so-called subnormal numbers. That is, whenever the exponent consists only of zeros, there is no implicit 1 in the mantissa, but the exponent is higher by one. That is, the exponents represented by bits 000 and 001 both correspond to $-2$. If our 8 bit toy arithmetic also used subnormal numbers, then 0000 0000 would be 0 and 000 0001 would be $(0 \cdot 1 + 0 \cdot 0.5 + 0 \cdot 0.25 + 0 \cdot 0.125 + 1 \cdot 0.0625) \cdot 2^{-2} = 0.015625$. We note that there is a positive and a negative zero (and also inf).

Similarly, floating-point numbers whose exponents consist only of ones are special. If additionally the mantissa is all zeros, then it represents inf and if the mantissa contains a non-zero bit, then it represents not-a-number (NaN) and the set bits correspond to error messages. ¨

### C.2  OPERATIONS WITH FLOATING-POINT NUMBERS

To distinguish the mathematical operations (infinite precision) from the computer arithmetic ones, we will use $\oplus, \ominus$ instead of $+, -$ to represent floating-point operations. When writing, e.g., $5 \oplus 7 = 12$, we mean that the floating-point representation of $5$ added to the floating-point representation of $7$ results in a floating-point $12$. We also note that $a \oplus -b = a \ominus b$.

The addition (or analogically subtraction) of two floating-point numbers is performed in three steps. First, the number with the lower exponent is transformed to the higher exponent; then the addition is performed (we assume with infinite precision), and then the result is rounded to fit into the floating-point representation. The standard allows for several rounding schemes, but the common one is to round to the closest number breaking the ties by rounding to the number with mantissa ending with 0.

For the sake of completeness, we also mention the multiplication of floating-point numbers. The multiplication is done in a way that exponents are added; the mantissas are multiplied and consequently normalized. We will not use (nontrivial) floating-point multiplication in our constructions.

Let us show an example of the floating-point addition.

**Example C.2.** Consider the addition of binary numbers, 1110 1010, and 0101 0011. The first one we already decoded as $-1.1010 \times 2^3$ and the other one is $1.0011 \times 2^2$; both in base 2.

$$1110\ 1010 \oplus 0101\ 0011 = -1.1010 \times 2^3 + 1.0011 \times 2^2 = -1.1010 \times 2^3 + 0.10011 \times 2^3,$$
$$= -1.00001 \times 2^3 \approx -1.0000 \times 2^3 = 1110\ 0000.$$

In base 10, we would have $-13 \oplus 4.75 = -8$ due to the loss of the least significant bits. This happened even though the exponents were different by the smallest possible difference. Consider further $6.5 \oplus 4.75 = 11$; Here, the loss of precision appeared even with equal exponents.

Unsurprisingly, it also holds that $6.5 \oplus 4.5 = 11$. Therefore, the addition to $6.5$ is not injective and, as a consequence, it is not surjective. Connecting this to randomized smoothing, we know that there are numbers which cannot be smoothed from $6.5$ as the following example shows.

**Example C.3.** When observing $2.125$, it could not arise as $6.5 \oplus a$ for any $a$. Indeed, $6.5 \oplus -4.5 = 2$, while $6.5 \oplus -4.25 = 2.25$. The representations are: $6.5 \sim 0101\ 1010$, $2.125 \sim 0100\ 0001$, $-4.25 \sim 1101\ 0001$ and $-4.5 \sim 1101\ 0010$. Here $-4.25$ is the smallest number bigger than $-4.5$. Note again that the exponents of $6.5$ and $2.25$ differ only by the smallest possible difference.

Another consequence is that floating-point addition is not associative. That is, the following identity does not always hold $(a \oplus b) \oplus c = a \oplus (b \oplus c)$.

**Example C.4.** Consider the numbers $a = 2.375 \sim 0100\ 0011$, $b = 3.75 \sim 0100\ 1110$, and $c = 3.25 \sim 0100\ 1010$. Then $a \oplus b = 6 \sim 0101\ 1000$ and $(a \oplus b) \oplus c = 9 \sim 0110\ 0010$. On the other hand, $b \oplus c = 7 \sim 0101\ 1100$ and $a \oplus (b \oplus c) = 9.5 \sim 0110\ 0011$; thus, the triple $a, b, c$ serves as a counterexample for associativity of $\oplus$.

## D  PROOF OF PROPOSITION 4.1

**Proposition D.1.** *There is a classifier with certified robust accuracy* $100\%$ *on the first* $1000$ *CIFAR10 test set images* $X \subset [0, \frac{1}{255}, \ldots, 1]^{3072}$ *(where we define class* $0$ *to include classes* $0, 1, 2, 3, 4$ *of CIFAR10 and class* $1$ *contains the other classes) with* $\ell_2$*-robust radius of* $3$ *and failure probability* $0.001$ *using randomized smoothing certificates, while for every point* $x \in X$ *there is an adversarial example* $x'$ *with* $\|x - x'\|_2 \le 1$.

*Proof.* We take $X_0 \subseteq X$ to be the set of all images from $X$ with class $0$ and $X_1 = X \setminus X_0$. Then we construct a hard classifier

$$M(x) = \begin{cases} 1 & \text{if } H_{X_1}(x) = 1 \text{ or } (H_{X_0}(x) = 0 \text{ and } x_1 > \frac{127}{255}), \\ 0 & \text{otherwise}, \end{cases}$$

where we use $H_A$ from (4). Experimentally, we conclude that for the smoothed classifier of $M$ with $\sigma = 1$, randomized smoothing certifies robust radius $3$ in $\ell_2$ norm for every point $x$ of the test set. At the same time, the perturbation $p = (\alpha \cdot \frac{240}{255}, \frac{-1}{255}, \frac{-1}{255}, \ldots) \in \mathbb{R}^{3072}$, where $\alpha$ is $1$ when looking for adversarial perturbation of class $0$ and $-1$ otherwise are universal adversarial perturbations. It holds for $x \in X$ that $\hat{M}(x)$ is correct; $\hat{M}(x) \ne \hat{M}(x + p)$, and also $\|p\|_2 \le 1$. □

## E  CERTIFICATION OF REAL-VALUED INPUTS OR DIFFERENT QUANTIZATIONS

Here we shall discuss the adaptation of the method to real-valued inputs, and also to other quantization levels.

### E.1 GENERALIZATION TO OTHER QUANTIZATION LEVELS

We described the method assuming 256 quantization levels and later we rounded the input to exact the same levels. This choice was arbitrary. The motivation was that the more fine-grained the levels are, the more similar the less changes will the rounding introduce. However, when choosing the quantization level after rounding, one should have in mind that the efficiency of the method relies on Proposition 5.1. It is easy to see that in general, we need to compute approximately $\text{lcm}(q_{in}, q_{out})$ integrals to be able to perform sampling, when $q_{in}$, $q_{out}$ are the respective inverse distances between neighbouring quantization levels of input, and after rounding. E.g., If we decided to round the input on scale $1/256$ instead of $1/255$, we would need to compute 256 times more integrals than when the quantization with the current rounding.

### E.2 GENERALIZATION TO REAL-VALUED INPUTS

Here we propose two potential generalizations of the method to real-valued inputs. The first produces maximal (in the sense of NP lemma) certificates at the cost of slow execution. The other brings no slow down, but the certificates will not be maximal.

As discussed in the previous subsection, Proposition 5.1 is crucial for the efficiency. However, with the real-valued inputs, it cannot be taken advantage of, because the inputs will (in general) be arbitrarily distant from 0. Thus, we would need to compute the samples from $\mathcal{N}_D^k(x, \sigma^2 I_d)$ independently for every input. While it is not a fundamental problem and we, in principle, can do so, it will become slow for high-dimensional images. To soften this problem, we can decide to have small number of quantization levels (e.g., 2) so we would need to compute just a single integral per input dimension. Using 2 quantization levels was already considered in Levine and Feizi (2021) and it yields the state-of-the-art $\ell_1$ robustness.

Another possible solution is to round the input before the certification. Let $x \in \mathbb{R}^d$ be the input point, we chose a quantization grid and its closest element to $x$ is $x'$. Then if we certify robust radius $r$ around $x'$, it implies that the robust radius centered at $x$ is at least $r - \|x - x'\|$. For instance, considering CIFAR10 dimension $d = 3072$ and the distance between quantization points is $1/1000$, then $\|x - x'\|_2 \le \left(3072 * 1/2000^2\right)^{1/2} \approx 0.055$. At the same time, we know that the certified radius at $x'$ would be at-most $r + \|x - x'\|$. Thus, the error is controlled and not significant. We can use more fine-grained quantization to make this error even smaller.

Finally, we note that the main focus of this paper is on quantized input as used in image classification. We expect that there are more sophisticated solutions to this problem e.g., by combining both proposed variants with rejection sampling. We leave this to future work.

## F GETTING A-ROUND GUARANTEES: FLOATING-POINT ATTACKS ON CERTIFIED ROBUSTNESS

In this appendix we discuss the floating-point attacks on randomized smoothing certificates of Jin et al. (2022). We could not reproduce the result which could be due to the following problems:

- The certificates of randomized smoothing are w.r.t. the smoothed classifier which is impossible to evaluate and we approximate it by random sampling. Thus, if we certify robust radius $r$ at point $x$ for classifier $f$, then if some $x_1$, $\|x - x_1\| <= r$ should be considered as an adversarial example (with high probability), we should also ensure that $f(x_1) = f(x)$ with high probability. That is, from the certification procedure we know $f(x)$ with high probability, but to know $f(x_1)$ with high probability, one has to determine confidence intervals e.g. using Clopper-Pearson or Hoeffding's inequality. However, in the paper, on bottom of page 13, there is written: `Given an instance x, the smoothed classifier g runs the base classifier f on M noise corrupted instances of x, and returns the top class kA that has been predicted by f.` This suggests that only a majority vote is performed; but to claim that one has found with high probability an adversarial samples not only the majority vote has to be wrong but there needs a significant gap between wrong and correct

class. Otherwise the result might just be bad luck due to random sampling and not indicate the existence of an adversarial sample.

- During the attack, they iteratively "refine" the adversarial perturbation. Thus, they evaluate the smoothed classifier multiple times. Since the outputs are probabilistic, when one tries enough candidates, one should in principle (incorrectly) find a "high probability" adversarial example just because one will be "lucky" with the randomness. This seems not to be taken into account in their method.

If their method is indeed a successful attack on randomized smoothing in floating point arithmetic, then this just emphasizes the need for a fix, which is exactly what we propose in this paper and overcomes the possibility of such an attack.

## G INTEGER-ARITHMETIC-ONLY CERTIFIED ROBUSTNESS FOR QUANTIZED NEURAL NETWORKS

Here we describe why the technique of Lin et al. (2021) for sampling from the discrete normal distribution and the consequent certification is not sufficient for our purposes.

The definition of the discrete normal distribution from Lin et al. (2021) (coinciding with the one in Canonne et al. (2020)) is as follows:

$$\mathbb{P}_{x \in \mathcal{N}_H(\mu, \sigma^2)}[\![x = a]\!] = Z e^{-\frac{(a-\mu)^2}{2\sigma^2}},$$

where $Z$ is an appropriate normalization constant and the distribution is supported on the set of integers. For the certification, similarly to the standard smoothing, first, the lower bound $p$ on the probability of the correct class for the smoothed classifier is estimated. Then, the robust radius is computed as $\sigma \Phi_{\mathcal{N}_H}^{-1}(p)$, where $\Phi_{\mathcal{N}_H}^{-1}$ is the inverse CDF of discrete Gaussian. This can be seen at the very bottom of the second column on page 4 in Lin et al. (2021). Note, in Algorithm 1 of Lin et al. (2021) there is written only $\Phi^{-1}$, which according to the neighbouring discussions (and according to Thm 3.2 there) corresponds to $\Phi_{\mathcal{N}_H}^{-1}$. This certified radius is clearly not exact, because the possible certified radii can only be $\sigma$ multiples of the quantization levels because the smoothing distribution is discrete. For $\sigma = 1$, the possible robust radii are $0, 1, 2, \ldots$, while the actual robust radius may clearly be non-integral which makes sense even when considering quantized inputs; e.g., consider the perturbation $(1, 1, 0, \ldots)$ which has distance $\sqrt{2}$. Therefore, the smoothing as described in Lin et al. (2021) is restricted to certify only integer radii which is a significant restriction.

However, we were not able to verify the correctness of the method proposed in Lin et al. (2021). In the proof of Theorem 3.1, there is: "Notice that that $p_{c_A}^{lb} = \mathbb{P}[X \in \mathcal{S}_A]$, where $\mathcal{S}_A = \{z : \langle z - x, \delta \rangle \leq \sigma \|\delta\|_2 \Phi_{\mathcal{N}_{\mathbb{H}}}^{-1}(p_{C_A}^{lb})\}$". Where $p_{c_A}^{lb}$ is the lower bounded probability of the target class. However, since the smoothing distribution is discrete, the function $\Phi_{\mathcal{N}_{\mathbb{H}}}^{-1}$ is piecewise constant, therefore there are probabilities $p_1 \neq p_2$ with $\Phi_{\mathcal{N}_{\mathbb{H}}}^{-1}(p_1) = \Phi_{\mathcal{N}_{\mathbb{H}}}^{-1}(p_2)$, thus they will both generate the same set $\mathcal{S}_A$, but using the stated fact in the proof, it would yield $p_1 = \mathbb{P}[X \in \mathcal{S}_A] = p_2$, which is absurd. Since the (incorrect) fact ($p_{c_A}^{lb} = \mathbb{P}[X \in \mathcal{S}_A]$, where $\mathcal{S}_A = \{z : \langle z - x, \delta \rangle \leq \sigma \|\delta\|_2 \Phi_{\mathcal{N}_{\mathbb{H}}}^{-1}(p_{C_A}^{lb})\}$) is given without a proof, we cannot rely on the correctness of the method. Even assuming the correctness of the method, the produced certificates cannot be, as discussed above, exact and are only lower-bounds on the actual robust radius certifiable by randomized smoothing.

## H DISCUSSION ON PRESENTED MALICIOUS EXAMPLES

To reproduce the malicious examples from Section 4, it is important to carry every computation in the same precision. We tested it for both, single and double-precision, it likely holds also for the half-precision. If some calculations are done in single, and some in double precision, then the claimed results will probably not hold. Specially, if some calculation is performed in the single precision (e.g., transforming images from $\{0, 1, \ldots, 255\}$ to $\{0, 1/255, \ldots, 1\}$), casting it to double precision afterwards is not sufficient because the low mantissa bits are already lost. Although the codes are very simple, we enclose some of the snippets in the supplementary materials. Proposition 5.1 is verified

by a C++ program. We believe that for the demonstration it is sufficient to run only 100 noises per sample. With 100 000 samples, it is sufficient to observe 99 900 successes to claim robust radius 3 with probability 99.9%, and 50 500 successes to claim with probability 99.9% that the result is class 1. However, the runtime is about one day on a 64 core machine.

## I    HOEFFDING'S BOUND

**Proposition I.1** (Hoeffding's inequality). *Let $X_1, \ldots X_n$ be random variables with $\frac{1}{n}\mathbb{E}[\sum_{i=1}^n X_i] = \mu$ and $0 \le X_i \le 1$. Then it holds that*

$$\mathbb{P}\left(\left(\frac{1}{n}\sum_{i=1}^n X_i\right) - \mu \ge t\right) \le e^{-2t^2 n}.$$

*for any $t \ge 0$.*

We rewrite the inequality as

$$\mathbb{P}\left(\left(\frac{1}{n}\sum_{i=1}^n X_i\right) - t \ge \mu\right) \le e^{-2t^2 n}.$$

Now the lhs stands for the probability that when we subtract $t$ from the average, it will still be bigger than the mean. This is the failure case of randomized smoothing that we want to allow only with probability $\alpha = 0.001$. Thus, we want to compute $t$ - how much do we subtract from the average.

$$e^{-2t^2 n} = \alpha$$
$$-2t^2 n = \ln(\alpha)$$
$$t^2 = \frac{\ln(\alpha)}{-2n}$$
$$t = \sqrt{\frac{\ln(\alpha)}{-2n}}$$

## J    PSEUDO RANDOM NUMBERS

Here we discuss the issues regarding the random number generators. In reality, we don't have true random number generators, we only have pseudo-random number generators. In order to estimate the quantity $\hat{f}(x) = \mathbb{E}_{\varepsilon \sim \mathcal{N}(0, \sigma^2 I_d)} F(x + \varepsilon)$ with probabilistic guarantees, we need the actual random numbers. Otherwise, the probabilistic statement does not make sense (apart from trivial cases). Thus, a reasonable alternative is to require that no statistical test would distinguish in polynomial time between the generated pseudo-random numbers and the actual random numbers with non-negligibly better probability than chance. This property is guaranteed by so-called cryptographically secure random number generators (Yao, 1982).

## K    ALGORITHMS

Here we compare the actual algorithms of the standard randomised smoothing in Algorithm 1, and of the proposed method in Algorithm 2. The differences in the methods of the same name are highlighted by colors. The algorithms assume input to be in $\{0, 1/255, \ldots, 1\}$. We emphasize that our method is a simple extension (differences highlighted) of the standard randomized smoothing, where the two additional procedures in Algorithm 2 can be evaluated just once before the certification; thus, they do not slow down the method, neither it decreases the accuracy.

---

**Algorithm 1** Randomized smoothing certification of Cohen et al. (2019)

---

1: **procedure** SAMPLEUNDERNOISE($f, x, n, \sigma$)
2:     `counts` $\leftarrow [0, 0]$
3:     **for** $i \leftarrow 1, n$ **do**
4:         $\varepsilon \leftarrow \mathcal{N}(0, \sigma^2 I)$
5:         $x' \leftarrow x + \varepsilon$
6:         **if** $f(x') > 0.5$ **then**
7:             `counts`$[1] \leftarrow$ `counts`$[1] + 1$
8:         **else**
9:             `counts`$[0] \leftarrow$ `counts`$[0] + 1$
10:     **return** `counts`

11: **procedure** CERTIFY($f, \sigma, x, n_0, n, \alpha$)
12:     `counts0` $\leftarrow$ SAMPLEUNDERNOISE($f, x, n_0, \sigma$)
13:     $\hat{c}_A \leftarrow$ top index in `counts0`
14:     `counts` $\leftarrow$ SAMPLEUNDERNOISE($f, x, n, \sigma$)
15:     $p_A \leftarrow$ LOWERCONFBOUND(`counts`$[\hat{c}_A], n, 1 - \alpha$)
16:     **if** $p > \frac{1}{2}$ **then**
17:         **return** prediction $\hat{c}_A$ and radius $\sigma \Phi^{-1}(p)$
18:     **else**
19:         **return** ABSTAIN

---

---

**Algorithm 2** Sound randomized smoothing certification of $F \circ g_k$

---

1: **procedure** PRECOMPUTE ARRAY OF BREAKING POINTS$(k, \sigma^2)$    ▷ This function is evaluated only once
2:      `arr` $\leftarrow [0, \ldots, 0]$                                 ▷ Array of $2 \cdot 255 \cdot (k+1) + 1$ zeros
3:      **for** $i = -255k - 255, 255k + 254$ **do**
4:          `arr`$[255k - 255 + i] \leftarrow \lceil 2^{64} \int_{-\infty}^{(i+0.5)/255} \frac{1}{\sqrt{2\pi\sigma^2}} e^{-\frac{x^2}{2\sigma^2}} \, dx \rceil$
5:      `arr`$[2 \cdot 255 \cdot (k+1)] \leftarrow 2^{64}$
6:      **return** `arr`

7: **procedure** $\mathcal{N}_D^{k+1}(0, \sigma^2 I)$ ▷ This function is evaluated only on the first call with given arguments and the result is memorized. The relevant arguments are $k, \sigma$.
8:      `arr` $\leftarrow$ Precomputed array of breaking points for $\mathcal{N}_D^k(0, \sigma^2)$
9:      $\varepsilon \leftarrow [0, \ldots, 0]$                                         ▷ Array of $d$ zeros
10:      **for** $i \leftarrow 1, d$ **do**
11:          $t \leftarrow \mathbf{U}(0, 2^{64-1})$
12:          **for** $j \leftarrow -255(k+1), 255(k+1)$ **do**
13:              **if** `arr`$[j + 255k] = t$ **then**
14:                  **return** Failure
15:              **else if** `arr`$[j + k] > t$ **then**
16:                  $\varepsilon_i \leftarrow j/255$
17:                  **Break**
18:      **return** $\varepsilon$

19: **procedure** SAMPLEUNDERNOISE$(f, x, n, \sigma, k)$
20:      `counts` $\leftarrow [0, 0]$
21:      **for** $i \leftarrow 1, n$ **do**
22:          $\varepsilon \leftarrow \mathcal{N}_D^{k+1}(0, \sigma^2)$
23:          **if** $\varepsilon \neq$ Failure **then**
24:              $x' \leftarrow \max\{-k, \min\{k+1, x + \varepsilon\}\}$
25:              **if** $f(x') > 0.5$ **then**
26:                  `counts`$[1] \leftarrow$ `counts`$[1] + 1$
27:              **else**
28:                  `counts`$[0] \leftarrow$ `counts`$[0] + 1$
29:      **return** `counts`

30: **procedure** CERTIFY$(f, \sigma, x, n_0, n, \alpha, k)$
31:      `counts0` $\leftarrow$ SAMPLEUNDERNOISE$(f, x, n_0, \sigma, k)$
32:      $\hat{c}_A \leftarrow$ top index in `counts0`
33:      `counts` $\leftarrow$ SAMPLEUNDERNOISE$(f, x, n, \sigma, k)$
34:      $p_A \leftarrow$ LOWERCONFBOUND$($`counts`$[\hat{c}_A], n, 1 - \alpha)$
35:      **if** $p > \frac{1}{2}$ **then**
36:          **return** prediction $\hat{c}_A$ and radius $\sigma \Phi^{-1}(p)$
37:      **else**
38:          **return** ABSTAIN

---

