# OpenReview forum: "Sound Randomized Smoothing in Floating-Point Arithmetic"
_ICLR.cc/2023/Conference — ICLR 2023 poster_

### Official Review · Reviewer_rQaT · 2022-10-21

**Confidence:** 2
**Correctness:** 4
**Technical Novelty And Significance:** 2
**Empirical Novelty And Significance:** 3
**Recommendation:** 6

**Clarity, Quality, Novelty And Reproducibility:**

The paper is well-written and the experiments are reproducible.


**Strength And Weaknesses:**

The paper identifies qualitatively new issues with randomized smoothing approach when using floating-point arithmetic. One expects potentially more issues when the numbers that appear in a classifier are large as this when floating point makes large absolute error in representing numbers. Since randomized rounding is used as a model for understanding the accumulation of errors in floating-point arithmetic when analyzing algorithms, I suspect that the issues that the paper is raising are quite minor in practice.

The sound randomized smoothing procedure in floating-point arithmetic results in negligible computational overheads in the image classification example, when compared to unsound certificates.

A weakness of the paper is that the differences between the unsound practice and sound certificates are very small in the image classification example. Therefore, the manuscript has not demonstrated that caring about floating point arithmetic will make significant difference in practice.

**Summary Of The Paper:**

The paper explains that randomized smoothing is no longer sound in floating-point arithmetic. The paper then proposes a more solid approach to randomized smoothing when using floating-point precision that yields sound certificates for image classifiers. Since adversarial attacks are of critical importance for security-related applications, certified robustness is an important topic. These certificates need to hold in finite precision.

**Summary Of The Review:**

The paper presents an interesting issue with current robustness certificates and proposes a new approach to randomized smoothing when using floating-point precision.

---

> ### Author Response · Authors · 2022-11-14
> **Response to rQaT**
>
> Thanks for the review!
>
> **A weakness of the paper is that the differences between the unsound practice and sound certificates are very small in the image classification example. Therefore, the manuscript has not demonstrated that caring about floating point arithmetic will make significant difference in practice.** \
> We don't consider it a weakness, quite the opposite as otherwise one of the currently most promising technique for certified robustness would potentially turn out to be useless. We don't argue that the networks we used in the evaluation produce wrong certificates; thus, the best result one can hope for is to get the same (modulo randomness) accuracies.
> We care about floating point errors because for certificates, there should be no question if they are actually correct or not. As of now, there is no consideration of how floating point errors affect randomized smoothing - we show the vulnerability and propose a fix essentially for free. If the reader draws a conclusion that they do not need to care for floating point errors, we disagree; still, the problem of the effect of floating point errors in randomized smoothing is resolved. This is by itself an important result.
> ​
>
> We also copy the response to R1 to a similar question here as we think it is relevant:  \
> We are considering certified robustness; thus, there should be no question if the model is actually robust. We believe that the it is beneficial to eliminate loopholes of verification methods before (and not after) they are exploited. In particular, since we point out how a malicious actor could exploit the loophole to fake non-existing robustness. As robustness certificates might be required in the future in safety-critical systems, it is in the interest of all of us that they are correct. We see it as a rather positive result, that we could not find directly cases where the certificates are clearly wrong. But this does not rule out that they exist. As our fix is basically for free, we don't see any reason not to use our fix.

---

### Official Review · Reviewer_YCXh · 2022-10-22

**Confidence:** 3
**Correctness:** 4
**Technical Novelty And Significance:** 3
**Empirical Novelty And Significance:** 3
**Recommendation:** 6

**Clarity, Quality, Novelty And Reproducibility:**

The paper shows that randomized smoothing in floating point arithmetic, which has not been an obvious attack angle.

Overall it is well written, but could be improved.

**Strength And Weaknesses:**

* Strength

The paper shows that randomized smoothing in floating point arithmetic, which has not been an obvious attack angle.

* Weakness

The proposed solution requires i.i.d. random numbers, which in practice we don't have. How would the paper modify the proposed solution if we know the random numbers are coming from a pseudo random generator?

**Summary Of The Paper:**

The paper shows that randomized smoothing in floating point arithmetic is susceptible to adversarial attacks.

**Summary Of The Review:**

I haven't seen many papers showing that floating point operations can be used to attack neural nets.

---

> ### Author Response · Authors · 2022-11-14
> **Response to YCXh**
>
> Thanks for the review! \
> **How would the paper modify the proposed solution if we know the random numbers are coming from a pseudo random generator?** \
> Thank you for the interesting question! The standard version of randomized smoothing (i.e., estimate the expected value by Monte Carlo sampling) requires random numbers. On the other hand, if we use a cryptographically secure pseudo random number generator (CSPRNG), then there is no polynomial (in the number of random bits used) time algorithm deciding (with non-negligible accuracy above chance) whether actual random numbers were used by the algorithm or not.  Thus, when using a CSPRNG, an attack on the certificates seems infeasible to us. We have added a discussion in the appendix L.
>
> ​
> **Overall it is well written, but could be improved.** \
> We would be happy to improve the writing, could you give us some suggestions please?
> ​

---

### Official Review · Reviewer_88Dn · 2022-10-25

**Confidence:** 4
**Correctness:** 4
**Technical Novelty And Significance:** 3
**Empirical Novelty And Significance:** 3
**Recommendation:** 8

**Clarity, Quality, Novelty And Reproducibility:**

The paper is very well written and due to the simplicity of the approach, it can be easily reproduced. Given the prior work on similar issues in differential privacy, the result is expected, but I believe there is value in bringing this to the ML community. Other than that, the only downside is the lack of an example beyond the toy example showcase.

Two questions regarding the reuse of noise:
Is there any upside to using different noise for each sample rather than the same precomputed one?
If so, would it be possible to create a large database of discrete noise samples and then just sample the required number from these for each sample?

Minor Formatting Issues:
- citep is often used in text, when citet should be used.
- Table 1 is located far from where it is referenced.

**Strength And Weaknesses:**

**Strengths**:
- The presented problem is significant and relevant as many papers in the robustness space rely on randomized smoothing.
- The proposed solution seems to work well.
- The paper is well-written and easy to follow.
- The idea of reusing the same noise took me by surprise, but it does seem sound.


**Weaknesses**:
- No example on a real dataset/classifier.



**Summary Of The Paper:**

The paper discusses how implementations of Randomized Smoothing (RS), an algorithm for certified robustness, can become unsound due to floating point arithmetic.
RS performs multiple model evaluations under noise to determine a robust output. The finite precision of IEEE-754 floating point numbers causes the addition of noise not to work as mathematically expected. This leads to an overestimation of the robustness properties and unsound certificates. This is also showcased in a toy example.
Lastly, a solution based on discretized noise is proposed and evaluated.

**Summary Of The Review:**

The paper showcases an important issue of the popular randomized smoothing algorithm.
It is well-written and also proposed a simple yet effective solution.
Thus my recommendation is acceptance.

---

> ### Author Response · Authors · 2022-11-14
> **Response to 88Dn**
>
> Thanks a lot for your review and detailed comments! We have fixed the citep/citet problems in the updated version and we will remember to choose the best place for the table in the finale version.
>
> **Is there any upside to using different noise for each sample rather than the same precomputed one?** \
> The guarantees with reusing noise are per-sample (as considered in the literature). However, if one would like to have distributional guarantees (e.g.,  "with probability at least P, the following holds: At least A% of the inputs can be certified with robust radius R with failure probability no more than Q" for some P,A,R,Q), then we would need to sample the noise independently.
> ​
>
> **If so, would it be possible to create a large database of discrete noise samples and then just sample the required number from these for each sample?** \
> Technically yes. If the goal is to evaluate a lot of models requiring independent failure cases. One only needs to be careful of memory consumption which will be hundreds of GB for certifying $500$ Cifar10 images ($100'000$ noises per image) and terabytes for $1000$ ImageNet images ($10'000$ noises per sample). If the goal is not to certify only a fixed benchmark, but do In the actual model deployment it is not clear how many samples would be needed.
> ​
>
> **No example on a real dataset/classifier.** \
> We briefly comment on the lack of example on a real classifier (we have examples with toy classifier on a real datasets, Propo 4.1). We agree that the message would be stronger, however we believe the loophole should be fixed before (and not after) it can be exploited.
> ​

---

> > ### Comment · Reviewer_88Dn · 2022-11-18
> > **Reply**
> >
> > Thank you for the update and the clarification on the noise independence.

---

### Official Review · Reviewer_9zy2 · 2022-10-25

**Confidence:** 3
**Correctness:** 4
**Technical Novelty And Significance:** 2
**Empirical Novelty And Significance:** Not applicable
**Recommendation:** 5

**Clarity, Quality, Novelty And Reproducibility:**

The paper points out and solves a new and important problem. It is very well-written, easy to follow, and provides enough details to replicate the experiments.

**Strength And Weaknesses:**

Strengths
- The paper closely analyzes the failure case for randomized smoothing certification under floating point errors. Provides a theoretical example where an image classifier might give false certificates because of numerical errors.
- The proposed solution of using a discrete normal distribution is quite simple and elegant. The paper also provides an efficient way to sample from the distribution making the approach practical.

Weakness
- Although not stated explicitly, the general belief is the safety guarantees of different data points are independent of one another. The single sampling framework needs to give a union bound over all points to ensure there are not too many failure cases.
- The paper does not show the vulnerability of real-world networks to floating point errors. Although the theory indicates that the networks might be vulnerable in the worst-case scenario, it is unclear how often these scenarios happen in real life.

**Summary Of The Paper:**

This paper identifies a new floating point arithmetic-based problem with the randomized smoothing-based certification framework. The authors show that a point can have an erroneously big certificate while being vulnerable to attacks. The paper proposes two rounding-based alternatives to standard randomized smoothing to address the floating point error problems. The authors also show that the run time of the proposed methods, as well as the performance, is comparable to standard randomized smoothing.



**Summary Of The Review:**

The paper points out an important problem that has gone unnoticed till now. The paper provides some good insights to solve this problem, but it is not fully clear how widespread it is in real-world networks. Finally, although it is an important problem that needs to be solved, I feel this might not be the right venue for this paper.

---

> ### Author Response · Authors · 2022-11-14
> **Response to 9zy2**
>
> Thank you for your review! We are happy that you think that "The paper points out and solves a new and important problem. It is very well-written, easy to follow, and provides enough details to replicate the experiments."
>
> **Although not stated explicitly, the general belief is the safety guarantees of different data points are independent of one another. The single sampling framework needs to give a union bound over all points to ensure there are not too many failure cases.** \
> Please note that  we consider both dependent and independent failure cases. The independence of failure cases comes at the cost of increased certification time, see Table 1 and Appendix A.1. We agree that it might make sense to demand the independence of the failure cases - although it is not required in the literature; thus, we discussed and provided both versions.
>
> ​
> **The paper does not show the vulnerability of real-world networks to floating point errors. Although the theory indicates that the networks might be vulnerable in the worst-case scenario, it is unclear how often these scenarios happen in real life.** \
> We are considering certified robustness; thus, there should be no question if the model is actually robust. We believe that the it is beneficial to eliminate loopholes of verification methods before (and not after) they are exploited. In particular, since we point out how a malicious actor could exploit the loophole to fake non-existing robustness. As robustness certificates might be required in the future in safety-critical systems, it is in the interest of all of us that they are correct. We see it as a rather positive result, that we could not find directly cases where the certificates are clearly wrong. But this does not rule out that they exist. As our fix is basically for free, we don't see any reason not to use our fix.
> ​
>
> **I feel this might not be the right venue for this paper.** \
> The manuscript addresses a problem of randomized smoothing for certified robustness. The common venues for papers in this area are ICML, ICLR and NeurIPS, we believe that this is indeed the appropriate venue and we think that such kind of phenomena are interesting and should be known to the ML community.

---

### Decision · Program_Chairs · 2023-01-20

**Decision:**

Accept: poster

**Justification For Why Not Higher Score:**

The authors’ results only apply under an idealised theoretical model of roundoff error.


**Justification For Why Not Lower Score:**

The paper makes an interesting contribution and would bring attention to the issue of roundoff error in certified robustness for randomised smoothing.


**Metareview: Summary, Strengths And Weaknesses:**

The authors demonstrate that robustness certificates computed for randomised smoothing can be rendered invalid due to floating point error. Under a theoretical model of floating point roundoff error, the authors develop a fix that enables computing valid certificates.

Strengths:
1. The authors identify a potential problem that invalidates a state of the art robustness certification technique. Prior work has only identified such issues for deterministic certification techniques, the authors show this issue arises in randomised smoothing as well
2. The authors develop a fix that enables computing valid certificates under a theoretical model of roundoff error.
Weaknesses:
1. The authors do not show a practical example (a real SOTA randomised smoothing classifier on a benchmark prediction task on MNIST/CIFAR10/ImageNet) where the roundoff error makes certificate computed for randomised smoothing invalid.
2. The fix developed by the authors only applies under an idealised theoretical model of roundoff error, not for the IEEE Floating Point Standard that defines roundoff error on real computing platforms.


**Note From Pc:**

if the above contains the word "oral" or "spotlight" please see: "oral" presentation means -> notable-top-5% and "spotlight" means -> notable-top-25%. As stated in our emails, we are disassociating presentation type from AC recommendations

**Summary Of Ac-Reviewer Meeting:**

The primary outcome of the discussion was to agree on the reasons to accept/reject. Reviewers were of mixed opinion about the paper but we agreed on the pros and cons of the paper. As an AC, I made a determination that the paper is worthy of acceptance as it will bring attention to this problem, and potentially trigger subsequent work that studies this problem under practical models of roundoff error.

Summary of discussion:

Points favoring acceptance:

The paper raises an important concern around the invalidation of certificates obtained via randomized smoothing, a SOTA certified robustness method.
The paper develops, under a theoretical model of floating point errors, a fix that enables obtaining valid certificates.
Points favoring rejection:

The paper is based on a theoretical model of floating point errors that do not conform to the IEEE floating point standard.
The paper does not present an actual practical example of a real network where the certificate is broken due to floating point errors.

---

> ### Author Response · Authors · 2023-02-28
> **We do not assume any rounding scheme.**
>
> We are thankful for the positive decision; however, we would like to emphasize that the proposed fix does not assume any rounding scheme because our method treats real numbers in a fully symbolic way and does not use floating point representations nor division at all; thus, the statement that the fix of our paper depends on a rounding scheme is wrong.
>
> Best,
> Authors